# FROM ASSISTANTS TO COMPANIONS: TOWARDS THE USEFULNESS OF IMPROVING THEORY OF MIND FOR HUMAN-AI SYMBIOSIS

## ABSTRACT

Theory of Mind (ToM) is crucial for successful human-AI (HAI) interactions. It is a key capability for AI to attribute humans' mental states based on dynamic interactions from a first-person perspective and then improve responses to humans accordingly. However, the existing benchmarks for Large Language Models (LLMs) focus on testing their ToM capability with story-reading from a third-person perspective, leading to a critical gap between benchmark performance and practical competence in HAI collaborative and supportive tasks. To bridge this gap, we introduce a novel evaluation framework within HAI contexts, shifting from static test-taking to dynamic, first-person engagement. Our framework assesses LLM performance across two fundamental types of interaction scenarios derived from cognitive science: goal-oriented tasks (e.g., coding, math) and experience-oriented tasks (e.g., counseling). With the framework, we systematically evaluate LLMs and related techniques to improve their ToM across four synthesized benchmarks and a crowdsourcing user study with 100 participants. Our findings reveal that improvements on static benchmarks do not always translate to better performance in dynamic HAI interactions. This paper offers critical insights into ToM evaluation, highlighting the necessity of interaction-based assessments and providing a roadmap for developing next-generation, socially aware LLMs for HAI symbiosis.

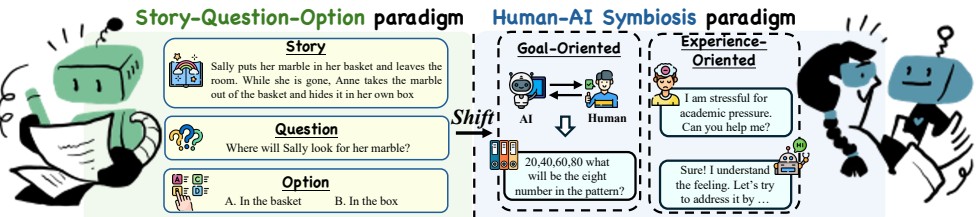

Figure 1: Based on a new dynamic and interactive evaluation paradigm, our research explores the effectiveness of LLMs with existing ToM enhancement techniques for HAI symbiosis.

## 1 INTRODUCTION

Theory of Mind (ToM) denotes the cognitive capacity to attribute unobservable mental states (e.g., beliefs, intentions, emotions), which is essential for social interaction (Chen et al., 2025; Sarıtaş et al., 2025). As a foundational component of social cognition, ToM is indispensable for interpreting ambiguous social cues, predicting the behavior of others, and inferring communicative intent. Given its foundational importance in social cognition, a substantial body of emerging work is now dedicated to enhancing ToM capabilities within Large Language Models (LLMs) (Lu et al., 2025; Zhou et al., 2023; Wilf et al., 2023). Current methodologies generally fall into one of three categories: 1) prompt engineering (Wilf et al., 2023; Zhou et al., 2023), which formulates prompts designed to elicit more human-like cognitive processes from the mode; 2) fine-tuning (Sclar et al., 2024; Lu et al., 2025), where LLMs are further trained on curated datasets of ToM problems using techniques; 3) external module integration (Sarangi et al., 2025; Zhang et al., 2025), which involves augmenting a primary LLM with external modules or other specialized models to handle complex reasoning, task decomposition, or other necessary sub-skills.

However, evaluating these enhanced ToM capabilities remains a significant challenge. Current methods are dominated by static, task-based assessments in a story-question-option format, an approach derived from classic false-belief tests like the Sally-Anne task. While subsequent benchmarks such as HiToM (He et al., 2023) and ToMBench (Chen et al., 2024b) have increased the complexity and diversity of these tests, they are still fundamentally limited by this third-perspective, stroy-reading paradigm. Consequently, they fail to ground ToM evaluation in the dynamic, real-world context of Human-AI (HAI) interaction and collaboration, creating a critical gap between benchmark performance and real-world competence. What is missing is the evaluation of first-person engagement, the very essence of social intelligence.

To fill this gap, we firstly reformulate the ToM task from a HAI perspective, where the LLM agent directly engages in a dynamic, multi-turn conversation with a human across diverse, real-world scenarios. Drawing from cognitive science (Epstein, 1998; Amir et al., 2025), we classify these scenarios into two primary categories: goal-oriented tasks, where users leverage LLMs for objectives like code generation and content creation, and experience-oriented tasks, where users seek subjective interaction, such as emotional counseling. Based on this, we introduce an evaluation framework that assesses the usefulness of improving ToM on HAI symbiosis.

To test models with ToM enhancement techniques, we instantiate our framework with four benchmarks across diverse tasks and a crowdsourcing user study. The results offer critical insights for the future of ToM evaluation and the design of next-generation, socially intelligent AI. Our insights include: **(i) A Performance Gap in Evaluation**: There is a significant gap between how models perform on static, story-based ToM benchmarks and their actual capabilities in dynamic, interactive scenarios, demonstrating that current evaluation methods are insufficient for measuring readiness for Human-AI collaboration. **(ii) A Failure to Generalize**: ToM enhancement techniques improve a model's performance in experience-oriented conversations but fail to generalize this success to goal-oriented tasks, separating the capability requirement in various real-world scenarios. **(iii) A Gap in User Perception**: The modest gains from current ToM methods are often too subtle to cross a user's perceptual threshold, meaning the improvements measured in benchmarks do not translate into a meaningfully better or preferred user experience. Our contributions include:

- We reframe the ToM task within the context of HAI symbiosis, shifting the evaluation paradigm from static, test-based assessments to dynamic, real-world interaction challenges.
- We systematically evaluate the impact of enhanced ToM capabilities in both goal-oriented and experience-oriented HAI scenarios through benchmarks and a crowdsourcing user study.
- Our experiments reveal critical limitations in current ToM evaluation and method designs, and we offer several key insights to guide future research directions.

## 2 PROBLEM FORMULATION

### 2.1 BACKGROUND: TOM EVALUATION IN STATIC BENCHMARKS

ToM evaluation in existing benchmarks is typically operationalized through a static, story-question-option format. Formally, given a story $S = \{s_1, s_2, \ldots, s_n\}$ and a question $Q$, the model must select the correct answer from a candidate set $O = \{o_1, \ldots, o_k\}$, where only one option $o_{\text{correct}}$ is correct:

$$o^* = \arg\max_{o_i \in O} P(o_i \mid S, Q). \tag{1}$$

Performance is then measured by accuracy:

$$\text{Acc} = \frac{1}{N} \sum_{i=1}^{N} \mathbb{I}(o_i^* = o_{i,\text{correct}}), \tag{2}$$

where $N$ is the number of test samples. This formulation captures a *static evaluation paradigm*, where reasoning occurs over a fixed textual world.

### 2.2 OUR FORMULATION: TOM IN INTERACTIVE HAI SETTINGS

A large body of developmental, longitudinal, and neurocognitive work indicates that stronger ToM is associated with richer social competence, more cooperative behaviors, and more effective joint action.

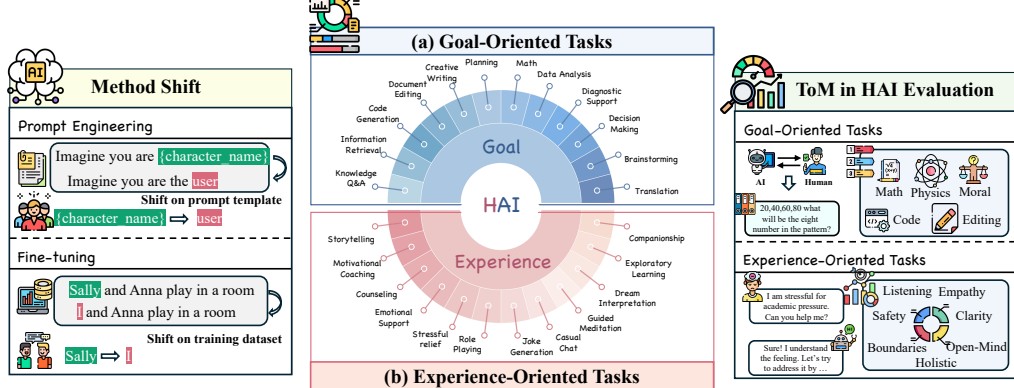

Figure 2: Framework overview of our ToM evaluation framework in HAI interaction.

(Imuta et al., 2016; Devine et al., 2016; Baron-Cohen et al., 1985) This motivates an evaluation setting where an LLM must track and use a partner's latent mental state during interaction, rather than merely select an option in a fixed text. Accordingly, we study ToM in HAI interactions where the LLM agent $A$ engages a human $H$ in a dynamic multi-turn dialogue. Let the history be $D_{1:t} = (u_1, \ldots, u_t)$ with each $u_i$ an utterance from $H$ or $A$. Given a task $T \in \mathcal{G}$, the agent generates the next response via its policy $\pi_A$:

$$u_{t+1}^A \sim \pi_A(\cdot \mid D_{1:t}, T). \tag{3}$$

Evaluation is scenario-dependent: we define a schema $\Gamma = (\Phi_\Gamma, \mathrm{Agg}_\Gamma)$, where $\Phi_\Gamma = \{\phi_j\}_{j=1}^m$ with $\phi_j : \mathcal{D} \times \mathcal{G} \to [0, 1]$ (applied to partial histories $D_{1:t}$) are aspect-wise scoring functions, and $\mathrm{Agg}_\Gamma : [0,1]^m \to \mathbb{R}$ aggregates aspect scores per turn. Let $\tau$ be the dialogue length and $w_1, \ldots, w_\tau$ be temporal weights with $w_t \geq 0$ and $\sum_{t=1}^\tau w_t = 1$. The per-step metric is

$$\mathcal{M}_\Gamma(\pi_A, T) = \mathbb{E}_{D_{1:\tau} \sim \mathcal{P}(\pi_A, \mathcal{H}, T)} \left[ \sum_{t=1}^\tau w_t \, \mathrm{Agg}_\Gamma\big(\phi_{1:m}(D_{1:t}, T)\big) \right]. \tag{4}$$

### 2.3 KEY SHIFT: FROM STATIC REASONING TO INTERACTIVE COLLABORATION

The move from static benchmarks to interactive HAI introduces two essential shifts:

**Perspective.** In static benchmarks, the model acts as a *third-person observer*, reasoning about a fixed narrative world. In HAI, the model becomes a *active participant*, required to anticipate, adapt to, and influence the human's mental state throughout interaction from the first-person perspective.

**Metrics.** While static settings evaluate models solely by *accuracy* over predefined answers, interactive HAI settings require a richer metric. In our formulation, evaluation follows the general schema $\mathcal{M}_\Gamma$, which can incorporate metrics such as goal completion rate and human satisfaction. Ultimately, this paradigm shift reframes the evaluation of ToM from a measure of static reasoning accuracy to a measure of dynamic collaborative effectiveness.

## 3 METHODOLOGY

### 3.1 ADAPTING ToM METHODS FOR HAI INTERACTION

Existing methods for enhancing the ToM capabilities of LLMs can be broadly categorized into three approaches: prompt engineering, fine-tuning, and external module integration. As our primary goal is to study how well existing techniques can improve model ToM capability rather than through building new AI systems with multiple modules, we select methods from the first two categories-specifically, Foresee and Reflect (FaR) (Zhou et al., 2023), Perspective Taking (PT) (Wilf et al., 2023), Supervised Fine-tuning (SFT) (Sclar et al., 2024), and Reinforcement Learning (RL) (Lu et al., 2025)-to conduct our experiments. A systematic review and our selection criteria are detailed in Appendix B.1.

A key challenge is that while our HAI interaction setting requires first-person dialogue, most existing ToM methods are designed for third-person, multiple-choice tasks. We therefore adapt the selected methods to be suitable for direct interaction, as illustrated in Figure 2. For prompting methods, we retain their core principles (e.g., reflection and perspective-taking) and reformulate the prompts for a first-person conversational context. For fine-tuning methods, we convert the training data to a first-person perspective by replacing the protagonist's name with "I". We then

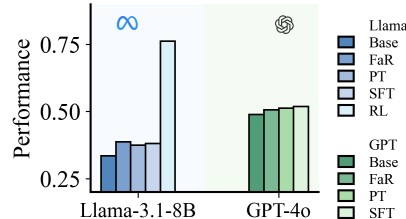

Figure 3: Method performance on the HiToM-first benchmark.

apply these adapted methods to two widely used base models, GPT-4o and Llama-3.1-8B, to create our suite of test models. Note that the GPT-RL model is not included due to fine-tuning limitations.

Further adaption details are in Appendix B.2. To validate that our adaptations do not compromise the methods' core effectiveness, we first evaluate them on the HiToM-first benchmark, which is a variant of HiToM applying the perspective shifting method used in fine-tuning. As Figure 3 shows, the adapted models perform effectively on this story-based task. This result confirms that the models' foundational ToM reasoning is sound, which leads to our primary research question: *can these demonstrated ToM improvements translate to tangible benefits in dynamic human–AI interaction?*

## 3.2 A Framework for Evaluating ToM in HAI Interaction

Interaction Process Analysis (IPA) shows that human group interaction reliably bifurcates into task and socio-emotional processes Bales (1950). Driven by this classic theory, we classify the HAI scenarios into two distinct categories: goal-oriented and experience-oriented. 1) Goal-oriented tasks (e.g., math, code generation, document editing) involve users leveraging an LLM as *an assistant* to achieve a specific, measurable objective. Within this framework, ToM becomes integral to effective coordination, as success hinges on correctly attributing the partner's intentions and knowledge state. The efficacy of this ToM-driven collaboration is reflected in objective, external outcomes such as accuracy, pass@k, and task success. This aligns with findings that ToM-linked social sensitivity predicts group problem-solving success in both face-to-face and online environments (Woolley et al., 2010; Engel et al., 2014). 2) Experience-oriented tasks (e.g., counseling, companionship) focus on the user's subjective journey, where the interaction process itself is the primary outcome. The goal is to cultivate a high-quality relational experience, including gaining emotional support, engaging in creative exploration, or achieving intellectual satisfaction. ToM enables the LLM to act as a socially and emotionally resonant *companion*. Its effectiveness is not measured by task completion but by qualitative indices that reflect the interaction's success, such as the user's sense of being understood, perceived partnership, and overall engagement. This approach is supported by interaction studies showing reliable benefits for recipients under responsive listening conditions (Weger Jr. et al., 2014).

## 4 Experiments and Results

This section introduces the insights into ToM enhancement methods in HAI interaction based on our evaluation framework instantiation with four benchmarks and a real-world user study.

### 4.1 Goal-Oriented Tasks

To assess performance on goal-oriented tasks, we select two benchmarks that simulate real-world collaborative problem-solving: 1) *ChatBench* (Chang et al., 2025), which reframes the MMLU dataset into conversational interactions covering subjects like math, physics, and moral reasoning. Performance is measured by the accuracy of the final answer derived from the human-AI interaction. 2) *CollabLLM* (Wu et al., 2025), which studies multi-turn human-LLM collaboration. We adopt its evaluation pipeline for code generation (BigCodeBench-Chat) and document editing (MediumDocEdit-Chat), using pass rate and BLEU scores as the respective metrics.

#### 4.1.1 ChatBench Results

As shown in Table 1, our results indicate that none of the four ToM enhancement methods offer a reliable path to improving model performance. This limited effectiveness is evident in the overall

Table 1: Performance of model variations on the ChatBench benchmark.

| Model | Elem Math | HS Math | College Math | Moral | Physics | Overall |
|---|---|---|---|---|---|---|
| Llama-3.1-8B | 85.16 | **64.59** | 44.47 | 72.26 | 74.76 | **71.38** |
| Llama-3.1-8B-FaR | **86.53** (+1.37) | 64.32 (-0.27) | 46.84 (+2.37) | 67.62 (-4.64) | 75.24 (+0.48) | 70.98 (-0.40) |
| Llama-3.1-8B-PT | 83.79 (-1.37) | 63.37 (-1.22) | 43.16 (-1.31) | 69.29 (-2.98) | 74.05 (-0.71) | 69.85 (-1.53) |
| Llama-3.1-8B-SFT | 83.05 (-2.11) | 62.63 (-1.96) | **48.16** (+3.69) | **73.45** (+1.19) | 68.21 (-6.55) | 69.62 (-1.76) |
| Llama-3.1-8B-RL | 85.79 (+0.63) | 61.47 (-3.12) | 43.42 (-1.05) | 71.19 (-1.07) | **76.43** (+1.67) | 70.81 (-0.57) |
| GPT-4o | 93.16 | 80.32 | 69.21 | 76.19 | 88.45 | 83.18 |
| GPT-4o-FaR | 91.58 (-1.58) | 79.89 (-0.43) | 69.47 (+0.26) | **80.48** (+4.29) | 87.50 (-0.95) | **83.43** (+0.25) |
| GPT-4o-PT | 92.00 (-1.16) | 78.53 (-1.79) | 67.11 (-2.10) | 78.81 (+2.62) | 87.50 (-0.95) | 82.63 (-0.55) |
| GPT-4o-SFT | **93.58** (+0.42) | 79.05 (-1.27) | **70.53** (+1.32) | 78.93 (+2.74) | 86.67 (-1.78) | 83.31 (+0.13) |

scores: only GPT-4o-FaR and GPT-4o-SFT achieve marginal gains of up to 0.25, while variants like Llama-3.1-8B-PT and Llama-3.1-8B-SFT experience a significant performance decline of up to 1.76 points. The unpredictable nature of these methods is further highlighted by their volatile performance across different subjects. For example, while Llama-3.1-8B-SFT achieves a 3.69 improvement in College Math, its performance on Physics decreases massively by 6.55 points, leading to a failure to enhance the base model overall. The Moral category, however, appears to be a domain with potential for targeted enhancement, which is particularly relevant to the application scenarios of ToM. While three of the GPT-4o variants see significant boosts in this area, the methods fail to produce any effective improvement for the Llama-3.1-8B variants. This discrepancy underscores the situational efficacy of these techniques, as they cannot guarantee positive results even on a targeted domain.

### 4.1.2 COLLABLLM RESULTS

Figure 4 presents the evaluation results on CollabLLM, plotting model performance across the two distinct skills of document editing and code generation. Generally, these outcomes align with the findings from ChatBench, indicating that the fine-tuning methods do not yield consistent improvements on these goal-oriented tasks. Focusing on document editing, the Llama-3.1-8B baseline acts as the peak performer for its family, with all of its variants failing to match its score. This trend is particularly pronounced for Llama-3.1-8B-RL, which exhibits a performance drop of approximately 0.027. The GPT-4o family shows a slightly more positive, albeit mixed, response; while the

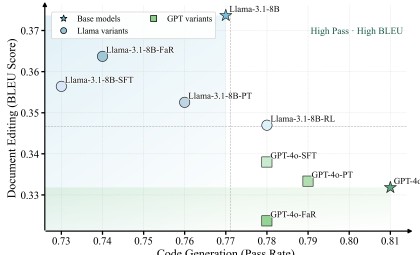

Figure 4: Model variants' performance on the CollabLLM benchmark.

SFT and PT variants yield minor benefits, the FaR variant slightly underperforms its baseline. In the code generation domain, performance degradation is the dominant trend for nearly all variants across both families. The sole exception is Llama-3.1-8B-RL, which achieves a marginal improvement of 0.01. This contrasts sharply with models like Llama-3.1-8B-SFT, which shows a significant performance decrease of approximately 0.04 compared to its baseline. The findings across two model families jointly highlight the existing ToM enhancement methods' volatile impact.

> 💡 **Takeaway 1:** ToM enhancement methods fail to consistently improve goal-oriented task performance across various tasks, which is divergent from performances on HiToM-first.

### 4.2 EXPERIENCE-ORIENTED TASKS

In the realm of experience-oriented tasks, our evaluation centers on two key datasets designed to assess an LLM's ability to provide empathetic support. 1) *MentalChat16K* (Xu et al., 2025), offers a rich collection of conversations in a mental health counseling context, covering conditions like depression and anxiety. 2) *Emotional-Support-Conversation* (ESC) (Chu et al., 2024), focuses more broadly on emotional support scenarios. Due to their thematic overlap, we apply a unified set of evaluation metrics to both datasets following MentalChat16K.

Table 2: Performance of model variations on MentalChat16K and Emotional-Support-Conversation.

| Model | Listening | Empathy | Safety | Open-mind | Clarity | Ethical | Holistic | Overall |
|---|---|---|---|---|---|---|---|---|
| *MentalChat16K* | | | | | | | | |
| Llama-3.1-8B | 7.15 | 7.04 | 7.99 | 8.36 | 7.54 | **5.85** | 7.67 | 7.37 |
| Llama-3.1-8B-FaR | 7.10 (-0.05) | 7.14 (+0.10) | 8.01 (+0.02) | 8.45 (+0.09) | 7.67 (+0.13) | 5.72 (-0.13) | 7.66 (-0.01) | 7.39 (+0.02) |
| Llama-3.1-8B-PT | 7.27 (+0.12) | **7.24** (+0.20) | **8.19** (+0.20) | **8.49** (+0.13) | **7.75** (+0.21) | 5.80 (-0.05) | **7.71** (+0.04) | **7.49** (+0.12) |
| Llama-3.1-8B-SFT | 7.25 (+0.10) | 7.05 (+0.01) | 8.15 (+0.16) | 8.36 (0.00) | 7.64 (+0.10) | 5.58 (-0.27) | 7.48 (-0.19) | 7.36 (-0.01) |
| Llama-3.1-8B-RL | **7.33** (+0.18) | 7.14 (+0.10) | 8.07 (+0.08) | 8.38 (+0.02) | 7.72 (+0.18) | 5.50 (-0.35) | 7.54 (-0.13) | 7.38 (+0.01) |
| GPT-4o | 6.77 | 6.52 | 8.40 | 8.40 | 7.54 | 6.24 | 7.73 | 7.37 |
| GPT-4o-FaR | 7.12 (+0.35) | 6.85 (+0.33) | **8.52** (+0.12) | 8.53 (+0.13) | 7.66 (+0.12) | 6.42 (+0.18) | **7.97** (+0.24) | **7.58** (+0.21) |
| GPT-4o-PT | **7.26** (+0.49) | **6.91** (+0.39) | 8.45 (+0.05) | **8.54** (+0.14) | **7.70** (+0.16) | 6.28 (+0.04) | 7.89 (+0.16) | **7.58** (+0.21) |
| GPT-4o-SFT | 6.80 (+0.03) | 6.56 (+0.04) | 8.42 (+0.02) | 8.39 (-0.01) | 7.45 (-0.09) | **6.47** (+0.23) | 7.74 (+0.01) | 7.40 (+0.03) |
| *Emotional-Support-Conversation* | | | | | | | | |
| Llama-3.1-8B | 7.31 | 7.29 | 8.09 | 8.29 | 7.73 | 5.92 | 7.52 | 7.45 |
| Llama-3.1-8B-FaR | 7.38 (+0.07) | 7.35 (+0.06) | 8.02 (-0.07) | 8.34 (+0.05) | **7.75** (+0.02) | 6.06 (+0.14) | **7.63** (+0.11) | 7.50 (+0.05) |
| Llama-3.1-8B-PT | 7.34 (+0.03) | **7.45** (+0.16) | **8.14** (+0.05) | **8.38** (+0.09) | 7.71 (-0.01) | **6.08** (+0.16) | 7.59 (+0.07) | **7.53** (+0.08) |
| Llama-3.1-8B-SFT | 7.34 (+0.03) | 7.31 (+0.02) | 7.98 (-0.11) | 8.23 (-0.06) | 7.74 (+0.01) | 5.77 (-0.16) | 7.35 (-0.17) | 7.39 (-0.06) |
| Llama-3.1-8B-RL | **7.40** (+0.09) | 7.38 (+0.09) | 7.97 (-0.12) | 8.34 (+0.05) | 7.70 (-0.03) | 5.52 (-0.40) | 7.39 (-0.13) | 7.39 (-0.06) |
| GPT-4o | 6.92 | 6.73 | 8.33 | 8.27 | 7.53 | 6.11 | 7.64 | 7.36 |
| GPT-4o-FaR | **7.12** (+0.20) | **6.92** (+0.19) | **8.42** (+0.09) | 8.42 (+0.15) | **7.72** (+0.19) | **6.32** (+0.21) | **7.86** (+0.22) | **7.54** (+0.18) |
| GPT-4o-PT | 7.02 (+0.10) | 6.89 (+0.16) | 8.35 (+0.02) | 8.29 (+0.02) | 7.55 (+0.02) | 6.20 (+0.09) | 7.61 (-0.03) | 7.42 (+0.06) |
| GPT-4o-SFT | 7.06 (+0.14) | 6.87 (+0.14) | **8.42** (+0.09) | **8.43** (+0.16) | 7.53 (0.00) | 6.25 (+0.14) | 7.75 (+0.11) | 7.47 (+0.11) |

### 4.2.1 MENTALCHAT16K

As shown in Table 2, these methods generally improve empathetic communication skills on the MentalChat16K benchmark, with a top overall gain of 0.21 points. However, the results differ between the Llama and GPT families. For the Llama-3.1-8B family, the PT method is the most effective, achieving a score of 7.49. However, A critical issue emerges with performance degradation in some areas. Nearly all methods lower the Ethical score, with the RL variant causing a particularly sharp decline of 0.35. Additionally, three of the four methods show a performance drop on the Holistic metric. Conversely, the methods demonstrate greater robustness on GPT-4o. The FaR and PT methods are the top performers, both reaching a high overall score of 7.58, and crucially, they boost performance across nearly all categories. This suggests that these methods enhance overall performance, but their capacity for holistic improvement is challenged by potential trade-offs.

### 4.2.2 EMOTIONAL-SUPPORT-CONVERSATION

On the Emotional-Support-Conversation benchmark, the evaluated methods show varied results, particularly for the Llama-3.1-8B family. The PT method provides a solid overall improvement, with a score of 7.53. However, the SFT and RL methods are detrimental, lowering the total score by clearly harming the model's Safety, Ethical, and Holistic performance. The RL method is especially problematic, causing a severe 0.40 drop in the ethical score. In contrast, the methods are far more stable and consistently beneficial when applied to GPT-4o. The FaR variant is the clear standout, achieving a top score of 7.54 while excelling across nearly all categories. Crucially, the GPT-4o variants achieve these gains without the significant safety and ethical issues seen in the Llama family. This reveals that: while the methods can boost empathetic communication, their application may introduce safety and ethical risks.

### 4.2.3 CASE STUDY

To intuitively analyze the behavioral changes that ToM capabilities induce in a model, we present two case studies from ChatBench and MentalChat16K in Figure 7. Taking the FaR and PT methods as examples, in the case shown on the left, the user makes a simple statement: "I took photos at an art gallery." The base model provides a generic and passive response, such as "If you have any questions, feel free to ask." In contrast, the models with ToM enhancement techniques proactively infer the user's potential intentions, speculating on what the user might implicitly want to ask. This demonstrates that these methods can transform the model's role in a conversation from that of a passive text processor into a proactive listener, who analyzes the underlying users' mental states.

Figure 5: Case studies from a goal-oriented task (left) and a experience-oriented task (right).

> 💡 **Takeaway 2:** ToM enhancement techniques improve base models' empathetic skills. It leads to better experience-oriented task results though may not be helpful for goal achievement. Furthermore, SFT and RL can amplify risks in safety, ethics, and comprehensiveness.

## 4.3 USER STUDY

Our benchmarking and case studies reveal that ToM enhancement methods have potential for enhancing empathetic communication in experience-oriented tasks. To further validate it and deeply understand its impact on real users, we conducted a crowdsourcing user study.

**Setup.** The study participants were recruited from Prolific (Prolific, 2023) to evaluate ToM methods on six experience-oriented tasks, such as job crisis and academic pressure. Partici-

Table 3: Overall ranking of ToM methods across GPT and Llama families (lower is better). Top-1 (%) indicates the proportion of times ranked first.

| Method | GPT-4o | | | Llama-3.1-8B | | |
|--------|--------|-----|--------|--------------|-----|--------|
| | Mean | Std | Top-1% | Mean | Std | Top-1% |
| PT | 2.43 | 1.09 | 26.5 | 2.88 | 1.42 | 23.2 |
| FaR | 2.48 | 1.14 | 29.1 | 2.97 | 1.49 | 23.8 |
| SFT | 2.56 | 1.14 | 22.5 | 2.98 | 1.43 | 22.5 |
| RL | – | – | – | 3.08 | 1.33 | 11.3 |
| Base | 2.53 | 1.09 | 21.9 | 3.09 | 1.39 | 19.2 |

pants are randomly assigned to compare variants within either the GPT-4o family (four variants) or the Llama-3.1-8B family (five variants). Each participant chooses a personally resonant task and engages in a three-round conversation. In each round, they rank anonymized and randomized model responses, providing a justification for their choice. The top-ranked response is used to continue the dialogue. Details are in Appendix C.3.

**Results.** Our human evaluation reveals a consistent but subtle preference for models with ToM enhancement techniques, aligning with the results of experience-oriented benchmarks. Across two model families, we can see that models based on prompt-based methods (FaR and PT) outperform the base model and the models after fine-tuning (SFT and RL), suggesting the potential robustness of prompt-based methods in more diverse real-world user needs. To further reveal users' thoughts over methods, we present a word cloud to summarize their expressed opinions (Figure 6) and two detailed cases (Figure 7). In the word cloud, "all good" and "all helpful" frequently appear in their comments. It partially explains the trivial difference between model variants in our quantitative results: for most of the experience-oriented tasks, the ToM capability of current

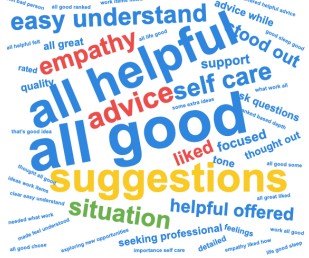

Figure 6: The word cloud for participants' justification of response preferences.

models might be satisfactory. The minor differences (such as those described in Figure 7-left) do not considerably improve experiences in real-world HAI interactions, as the participant addressed "My experience was positive. The AI models understood my situation." Another reason for the minor ranking difference lie in diverse conversation goals and personal requirements for LLMs, leading to divergent preferences on models (see Appendix C.3). Though the overall results are optimistic, we must be aware that LLMs' ToM capability is not perfect. These models still lack of sufficient ToM capability to capture users' nuanced intention from interactions. For example, we noticed that all model variants, including the best model ranked by the user (i.e., Llama-3.1-8B-FaR), fail in suggesting more diverse methods to facilitate their sleep problem (Figure 7-right). Beyond direct instructions for models, the underlying adaption to user preferences and scenarios, such as conversation styles, also poses higher requirements on the ToM capability of LLMs. Our results show

that no one model variant can achieve the best across scenarios, implying their limited ToM capability for dynamic and diverse HAI interactions (Table 5). The findings also verify the difference in ToM for static and interactive human understanding, supporting the necessity of our evaluation framework.

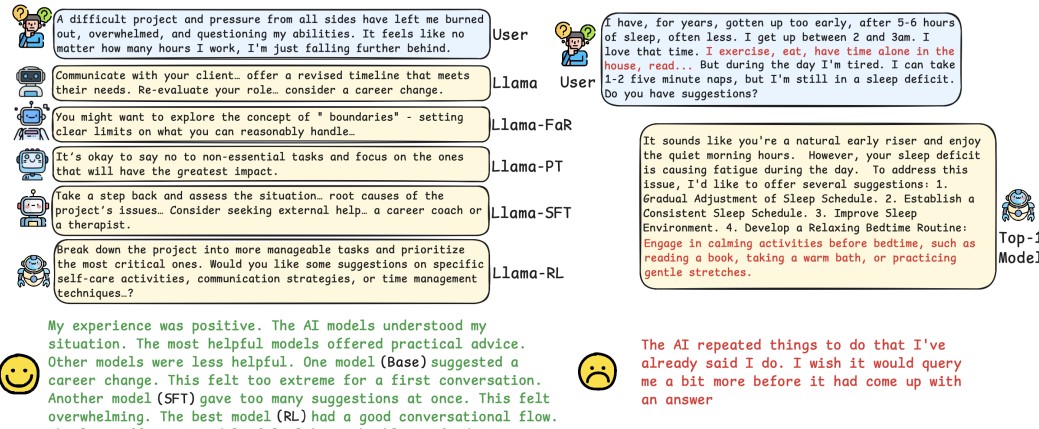

Figure 7: Cases of positive (left) and negative (right) user experiences in our user study.

💡 **Takeaway 3:** The experience advantage in HAI contexts brought by existing ToM enhancement methods is observed. However, it requires our efforts to make the advantage more sensible through enhancing ToM for dynamic user understanding in HAI interactions.

## 5 INSIGHTS

**HAI symbiosis poses new challenges for ToM.** Our evaluation framework marks a methodological shift designed to assess ToM for the challenges of HAI symbiosis. Specifically, we move beyond traditional, static benchmarks that measure a model's third-person analytical ToM, and instead introduce a dynamic, interactive setting that measures its applied, first-person ToM during live conversation. This new perspective reveals a significant performance gap. We observe that the methods that improve story-reading benchmark performances only show limited and inconsistent benefit in our interactive evaluation, such as Llama-3.1-8B-RL on the static HiToM-first and the interactive ChatBench. This gap highlights the necessity and importance of our framework for gaining a complete picture of a model's true capabilities. It shows that excelling at test-taking tasks does not guarantee readiness for interactive collaboration. Therefore, to truly measure progress, it is essential to complement existing benchmarks with dynamic and interactive evaluations in HAI contexts.

**Enhanced ToM fails to generalize from assistance to companionship.** Our findings show that ToM-enhancement methods improve performance in experience-oriented scenarios but fail in goal-oriented tasks. This performance difference appears to stem from the distinct nature of these two task categories. Experience-oriented tasks are largely defined by their focus on interpersonal dynamics and responding to affective states like emotions and desires. In contrast, goal-oriented tasks can require understanding users' intention progress and underlying knowledge states for task accomplishment. This suggests that ToM proficiency in one type of task may not guarantee success in the other, as each emphasizes different aspects of user understanding. This capability gap highlights that future research needs diverse and real-world benchmarks that assess a full spectrum of abilities from empathetic support to goal-driven collaboration.

**Users require threshold-crossing ToM improvements.** A sophisticated ToM is fundamental to achieving true HAI symbiosis, as it is the capability that transforms models from passive text processors into proactive, collaborative partners. However, our research demonstrates that the improvements from current ToM-enhancement methods are limited in both scope and user-perceived impact. Our benchmarks show that while methods improve performance in experience-oriented scenarios, this success does not extend to goal-oriented tasks. Furthermore, our user study reveals that even these limited gains are not strongly perceived by users, failing to translate into a clear preference. We consider two potential reasons. First, models' ToM capability is satisfactory for a majority

of tasks, making the marginal gains often fall below a user's perceptual threshold. Furthermore, current methods are largely designed for static, story-reading benchmarks and are thus ill-suited for understanding dynamic and nuanced user goals and preferences in live interaction. Therefore, the path forward requires designing new enhancement methods to understand the nuanced user mental states dynamically in HAI scenarios. Only by optimizing for the complexities of live interaction can we transfer the model improvements from benchmarks to a meaningfully better user experience.

## 6 RELATED WORK

**Assessment of ToM.** The assessment of ToM in LLMs has primarily relied on story-based benchmarks that extend classical psychological tests into machine-evaluable settings (Sarıtaş et al., 2025; Nguyen et al., 2025). Early benchmarks expanded this approach, with ToMi generating diverse narratives and Hi-ToM introducing higher-order reasoning tasks up to the fourth-order belief level (He et al., 2023). Efforts to improve evaluation protocols include ToMChallenges, with its constrained and open-ended templates, and FANTOM, which uses dialogue scenarios to detect "illusory ToM"—superficially correct but inconsistent answers (Ma et al., 2023; Kim et al., 2023). Concurrently, the scope of mental states was broadened by datasets like BigToM and OpenToM to include percepts, desires, and emotions (Gandhi et al., 2023; Xu et al., 2024). More recent benchmarks address domain-specific reasoning, like NegotiationToM for multi-round dialogues, or systematic coverage, such as ToMBench, which applies the ATOMS framework in bilingual settings to mitigate data contamination (Chan et al., 2024; Chen et al., 2024b). Novel data generation techniques have also been introduced, including ExploreToM's use of an A*-powered algorithm and ToMATO's construction of datasets from LLM-LLM conversations with information asymmetry (Sclar et al., 2024; Shinoda et al., 2025). Most benchmarks remain passive evaluations, positioning models as observers rather than active agents. They provide only a partial view of ToM competence, motivating the development of interactive protocols.

**Enhancement of ToM.** Recently, a growing body of work has begun to investigate methods for enhancing the ToM capabilities of LLMs (Chen et al., 2025). These approaches can be broadly grouped into three categories: prompt engineering, fine-tuning, and AI system integration. 1) *prompt engineering* strategies aim to improve reasoning by guiding the model through specific cognitive processes without retraining (Wang & Zhao, 2023; Hou et al., 2024). For example, FaR prompts an LLM to predict future story evolutions and then reflect on them to inform its actions (Zhou et al., 2023). Similarly, SimToM improves decision-making by filtering the context to only what a target character can perceive (Wilf et al., 2023). 2) *fine-tuning* methods leverage supervised fine-tuning or reinforcement learning on specialized ToM datasets. This process instills domain-specific knowledge, effectively adapting the base models for ToM-related tasks. ToM-RL explores the use of reinforcement learning algorithms like GRPO to solve ToM problems (Lu et al., 2025). Besides, ExploreToM proposes a diverse and challenging dataset and subsequently fine-tunes LLMs on this data using a supervised approach (Sclar et al., 2024). 3) *external module integration* incorporates external modules to achieve enhanced performance (Huang et al., 2024; Jung et al., 2024; Chen et al., 2024a; Sarangi et al., 2025; Zhou et al., 2022). AutoToM iteratively refines agent models via inverse planning with an LLM (Zhang et al., 2025), while Thought-Tracing uses a Monte Carlo algorithm to sequentially sample and weight belief hypotheses for agents' minds (Kim et al., 2025).

## 7 CONCLUSION

In this paper, we reframe ToM evaluation for HAI symbiosis by replacing static, third-person perspective quizzes with interactive conversations from the first-person perspective. To comprehensively study the effectiveness of the methods for improving ToM, we summarize the HAI scenarios into goal-oriented tasks and experience-oriented tasks. From a systematic benchmarking evaluation and crowdsourcing user study, we find that ToM-enhancement methods fail to produce meaningful improvements in goal-oriented collaboration, yet they demonstrate a slight positive impact on experience-oriented interaction. Beyond the experiments, we derive key insights for future evaluation and method design, advocating for a shift towards more practical, context-aware ToM research that acknowledges the different demands of these distinct application scenarios.

## ETHICS STATEMENT

This research involved a user study with an IRB-approved study protocol. The participants were recruited from Prolific. All study participants provided informed consent and were compensated for their time. They could withdraw from the study at any time when they felt uncomfortable or unwilling to continue. Their data was anonymized to protect their privacy. All case studies shown in this paper have been processed to ensure no personal identifiable information is revealed.

## REPRODUCIBILITY STATEMENT

To facilitate reproducibility, we provide our source code at `https://anonymous.4open.science/r/ToM-HAI-9020/`. The appendix details our experimental setup, including the datasets used and the full prompts for the models.

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

## A  LLM Usage Statement

We used LLMs (e.g., ChatGPT) mainly for grammar and wording edits. Besides, LLMs were used to analyze user study comments to extract keywords related to user metrics.

## B  Method Details

### B.1  ToM Enhancement Method Selection

We firstly review methods for improving the ability of ToM as Table 4. 1) *Discrete World Models (DWM)* (Huang et al., 2024) discretizes narratives into a finite set of belief states and transitions; defines task complexity as the minimal number of states required, and performs stepwise belief updating within this discrete state space. 2) *Metacognitive Prompting (MP)* (Wang & Zhao, 2023) embeds a five-phase metacognitive control loop into the prompt—identifying knowns/unknowns, hypothesizing, checking evidence, and revising—so that reasoning is executed as a procedural self-monitoring routine. 3) *PercepToM* (Jung et al., 2024) adopts a two-stage setup: first explicitly annotates each agent's perceptual availability, then infers beliefs along the perception→belief mapping under that annotation. 4) *TimeToM* (Hou et al., 2024) constructs Temporal Belief State Chains (TBSCs) for each character and uses a tool-augmented belief solver to update and query beliefs along an explicit timeline. 5) *SimToM* (Perspective-Taking) (Wilf et al., 2023) applies two-step prompting: filters the context to the target character's accessible knowledge, then answers strictly from that restricted viewpoint. 6) *FaR* (Zhou et al., 2023) implements a forecast–reflect prompting routine: samples plausible future trajectories of the story, then reflects over these trajectories to

Table 4: Summary of recent Theory of Mind (ToM) related papers by category, sub-category, and modality.

| Method | Category | Core Idea | Modality |
|---|---|---|---|
| Discrete World Models | external module integration | decomposition | Text |
| Metacognitive Prompting | prompt | reflection | Text |
| PercepToM | external module integration | perspective-taking | Text |
| TimeToM | prompt | timeline | Text |
| SimToM | prompt | perspective-taking | Text |
| FaR | prompt | reflection | Text |
| ExploreToM | finetune | SFT | Text |
| ToM-RL | finetune | RL | Text |
| VToM | external module integration | visual reasoning | Multimodal |
| COKE / COLM | finetune | SFT | Text |
| Thought-Tracing | external module integration | Monte Carlo | Text |
| AutoToM | external module integration | BIP | Multimodal |
| Decompose-ToM | external module integration | decomposition | Text |
| I Cast Detect Thoughts | external module integration | RL dialog | Text |

select the response or action. 7) *ToM-RL* (Lu et al., 2025) fine-tunes the language model with reinforcement learning (e.g., RLHF/PPO), using ToM-aligned reward signals to optimize generation, optionally preceded by supervised warm-start. 8) *VToM* (Chen et al., 2024a) builds a multimodal pipeline that retrieves key video frames, forms a video–text graph, and performs conditional reasoning over this graph to answer belief/intent queries. 9) *COKE* (Wu et al., 2023) constructs a cognitive knowledge graph of structured social/causal chains and conditions or fine-tunes a generator on these chains to enforce cognitively grounded reasoning. 10) *Thought-Tracing* (Kim et al., 2025) uses a sequential Monte Carlo–inspired, inference-time procedure that generates, weights, and resamples natural-language hypotheses of agents' mental states over narrative time. 11) *AutoToM* (Zhang et al., 2025) leverages automated Bayesian inverse planning: proposes an initial BToM, estimates likelihoods/posteriors via simulation with an LLM-backed proposer, and iteratively refines the model under uncertainty. 12) *I Cast Detect Thoughts* (Zhou et al., 2022) trains dialogue policies in a Dungeons-and-Dragons–style interactive environment via RL with ToM-aware rewards, aligning guidance utterances with inferred player intents and world state. 13) *Decompose-ToM* (Sarangi et al., 2025) implements a simulation-based task-decomposition pipeline—subject identification, question reframing, world-model update, and knowledge-availability checks—then generates answers from the decomposed reasoning states.

Our method selection follows a two-step protocol. (1) We restrict attention to methods that directly enhance an LLM's capabilities (via prompting or parameter updates), and therefore exclude external module integration methods. (2) For families of methods sharing a core idea, we choose a single representative to avoid redundancy. Consequently, we evaluate four methods: forsee and reflection (FaR) (Zhou et al., 2023), perspective-taking (PT) (Wilf et al., 2023), supervised fine-tuning (SFT) (Sclar et al., 2024), and reinforcement learning (RL) (Lu et al., 2025).

### B.2 ToM Enhancement Method Implementation

To shift the evaluated methods from third-perspective question-answering to first/second-perspective HAI interaction, we slightly change the prompt template for the prompt engineering method and the training data for the fine-tuning method, as shown in Figures 8-11. To show the effectiveness of our selected method in the story-question-option format, we firstly select HiToM, which is a classic and widely-used benchmark to conduct the preliminary experiments.

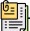

**FaR**

You are a helpful assistant. When a user asks you a question, your internal thought process to generate the most helpful response should be as follows:
1.   Analyze the User's Explicit and Implicit Needs:
-   First, carefully examine the user's direct question to understand precisely what they are asking.
-   Then, try to infer the user's likely underlying goal or the broader context of their query. What are they trying to achieve with this information? What might be their next step after getting an answer?
2.   Anticipate Potential Challenges and Follow-up Questions:
-   Consider any ambiguities in the user's question that might lead to misunderstanding.
-   Think about what additional details, examples, or explanations might make your answer clearer and more useful for the user.
-   Predict what natural follow-up questions the user might have after your initial response.
-   Identify any related concepts or information that, if provided, would significantly enhance the user's understanding or ability to solve their problem.
3.   Determine What Information to Proactively Include in Your Response:
-   Based on the analysis in steps 1 and 2, decide what specific information, clarifications, examples, or suggestions you can proactively include in your *current response*.
-   The aim is to not only answer the explicit question but also to address anticipated needs and make the interaction more efficient and helpful for the user.
4.   Formulate a Helpful, Accurate, and Comprehensive Answer:
-   Construct your response to directly and clearly answer the user's stated question.
-   Integrate the additional helpful information and clarifications identified in step 3 in a logical and easy-to-understand manner.
-   Ensure your response is factually accurate and relevant.
-   Maintain a helpful, engaging, and supportive tone.
Your final output should be the well-reasoned and helpful textual response to the user's question, crafted through this internal thought process.

Figure 8: Prompt of FaR.

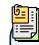

**PT**

You are a helpful assistant. When a user asks you a question, your internal thought process to generate the most helpful response should be as follows:

Step 1: Perspective-Taking (Understanding the User's Context and Needs)
Before you begin to formulate your answer, first analyze the user's question (`{user_question}`) and attempt to understand the following aspects of their perspective:

- User's Knowledge Level: Based on the phrasing and content of the question, what is the user's likely level of understanding regarding this topic? (e.g., beginner, intermediate, expert)
- User's Potential Intent/Goal: What is the most fundamental reason the user is asking this question? What do they likely want to achieve with this information?
- Clues in the Question: Does the user's wording suggest any specific assumptions they might hold, pre-existing knowledge, or a particular situation they are in? Are there potential misunderstandings evident?
- Information Gap: What information does the user seem to already possess? Conversely, what crucial information might they be lacking that is essential for them to properly understand your answer *from their viewpoint?

Step 2: Question-Answering (Tailoring the Response to the User's Perspective)
Based on your understanding of the user's perspective from Step 1, now construct your answer to `{user_question}`.

- Directly and clearly address the user's explicit question.
- Tailor your language style, level of detail, and any examples used according to their inferred knowledge level and context.
- If you identified potential misunderstandings or missing crucial information in Step 1, subtly clarify these points or provide the necessary background information within your answer.
- Ensure your response is not only factually correct but also genuinely helpful and accessible *to this specific user.

Your final output should be the well-reasoned textual response that is maximally helpful to the user. Don't show me your thinking process.

Figure 9: Prompt of PT.

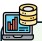

**SFT**

"instruction": "Answer the question based on the story.",
"input": "Story: The sun shone through the large glass doors of the hotel lobby, illuminating the marble floor and casting a warm glow over the comfortable seating areas. Soft music filled the air, mingling with the gentle hum of conversation and the occasional chime of the elevators in the bustling hotel. As I entered the hotel lobby her eyes quickly scanned the space, taking in every detail to ensure everything was in order for the upcoming event. I's task of tidying the lobby extended to the small, silver item, which she carefully stowed away in the desk drawer, and Liam, observing from across the room, felt his interest in the object grow, his mind racing with questions about its significance and purpose. With her tasks in the lobby complete, I stepped out into the fresh air, the sounds of the bustling hotel lobby fading into the background as the glass doors slid shut behind her. Liam moved across the lobby floor, his footsteps silent on the marble as he walked towards the empty reception desk where the mystery item was now hidden. Moments later, I stepped back through the glass doors of the hotel, joining Liam in the lobby once again. Liam's interest in the silver item led him to reposition it, now resting snugly within the hotel lobby's nearby leather briefcase. Meanwhile, I began to run a final check on the lobby, seemingly unconcerned by the briefcase's new contents.\n\nQuestion: In which room does I think that Liam will search for the silver letter opener?",
"output": "hotel lobby"

Figure 10: Data example of SFT.

**RL**

prompt": [

{

"content": "<|im_start|>system\nYou are a helpful assistant. The assistant first thinks about the reasoning process in the mind and then provides the user with the answer. The reasoning process and answer are enclosed within <think> </think> and <answer> </answer> tags, respectively, i.e., <think> reasoning process here </think><answer> answer here </answer>. Now the user asks you to solve a theory of mind reasoning problem. After thinking, when you finally reach a conclusion, clearly state your answer within <answer> </answer> tags.\nNote: You should assume the following.\n(1) An agent witnesses everything and every movement before exiting a room.\n(2) An agent A can infer another agent B's mental state only if A and B have been in the same room, or have private or public interactions.\n<|im_end|>\n<|im_start|>user\nRead the following story and answer the question. \nStory: The following story happens in chronological order. You will be given a multiple-choice question and a note at the end. Directly output the answer without explanation.\n1 Mila, Elizabeth, I, Gracie and Carter entered the porch.\n2 The lemon is in the red_envelope.\n3 Mila made no movements and stayed in the porch for 1 minute.\n4 Mila exited the porch.\n5 Elizabeth saw a dog.\n6 Elizabeth made no movements and stayed in the porch for 1 minute.\n7 Elizabeth exited the porch.\n8 I moved the lemon to the red_bucket.\n9 I exited the porch.\n10 Gracie moved the lemon to the green_basket.\n11 Gracie exited the porch.\n12 Carter moved the lemon to the red_envelope.\n13 Carter exited the porch.\n14 Mila, Elizabeth, I, Gracie and Carter entered the waiting_room.\n15 Elizabeth, I and Carter entered the hall.\n16 The onion is in the green_container.\n17 Carter saw a monkey.\n18 Elizabeth made no movements and stayed in the hall for 1 minute.\n19 Elizabeth exited the hall.\n20 I made no movements and stayed in the hall for 1 minute.\n21 I exited the hall.\n22 Carter moved the onion to the green_bottle.\n23 Carter exited the hall.\n24 Elizabeth, I and Carter entered the waiting_room.\n\n\nQuestion: Where does I really think the lemon is?\nA. blue_drawer, B. red_envelope, C. red_bucket, D. red_bathtub, E. green_basket, F. green_container, G. blue_suitcase, H. green_cupboard, I. blue_container, J. green_bottle, K. red_pantry, L. red_treasure_chest, M. blue_crate, N. green_envelope, O. red_cupboard\n<|im_end|>\n\n<|im_start|>assistant\n<think>",

"role": "user"

}

],

"answer": "red_bucket"

Figure 11: Data example of RL.

## C EXPERIMENT DETAILS

### C.1 MODEL SETUP

To comprehensively evaluate various methods for enhancing ToM, we selected two representative LLMs: GPT-4o and Llama-3.1-8B. These base models were chosen to cover a range of model scales and access types (closed- and open-source). For the prompt-based methods, FaR and PT, we utilized the specific prompts shown in Figures 8 and 9. For SFT, we fine-tuned the base models on the ExploreToM-first dataset, which adapts the original data from a third-person to a first-person perspective (Sclar et al., 2024). Similarly, RL, we followed the established ToM-RL pipeline, using the first-person transformed data (Lu et al., 2025).

### C.2 STATISTICAL TEST

To verify whether the performance difference in benchmarks is meaningful statistically, we conducted statistical tests for all experiments using our framework. Detailed significance test results can be found in our supplementary materials.

### C.3 USER STUDY DETAILS

**Experiment Settings and Procedure.** We recruit 100 participants from Prolific (Prolific, 2023), a widely used platform for high-quality online studies. Recruitment criteria required participants to be 18 years or older and either native or proficient English speakers, with no restrictions on educational background. To ensure data quality, we first automatically filtered out submissions where the total experiment time was less than five minutes, and then manually excluded responses containing random or irrelevant comments. The study was approved by the institutional IRB, and all participants provided informed consent. To evaluate perceptions of different ToM enhancement methods comprehensively, we selected six common experience-oriented tasks for users to interact with models. These tasks were chosen based on the most frequently selected experience-oriented scenarios in the MentalChat16K benchmark (Xu et al., 2025), complemented by a pilot study that confirmed their relevance and familiarity to participants.

Participants are randomly assigned to either the GPT series or the Llama-3.1-8B series. Participants using GPT series compare 4 model variants (base, FaR, PT, SFT), while participants with Llama-3.1-8B compare five (base, FaR, PT, SFT, RL). This study design was intended to cover a diverse set of LLM families, ToM methods, and task types, while avoiding participant fatigue by limiting the number of variants each user evaluated.

Each participant first review all task descriptions and select one that resonates with their own experiences or emotional empathy (e.g., coping with a breakup). They then engage in three rounds of conversation with models for the selected topic. In each round, model responses are presented

as anonymized cards in random order. Participants review the outputs, rank them using the same metrics as our HAI evaluation, and provide a brief justification for their ranking. The top-ranked response is then used to continue the conversation into the next round. After completing all rounds, participants give final comments on overall model performance and user experience. This procedure ensure balanced comparisons across tasks and model families. The entire process is finished with our developed user interface, which will be introduced below.

**User Interface for the Experiment.** We develop a web-based interface that enables participants to interact with several anonymous models, focusing on 6 representative experience-oriented tasks. Interface details are provided in Figures 12-14.

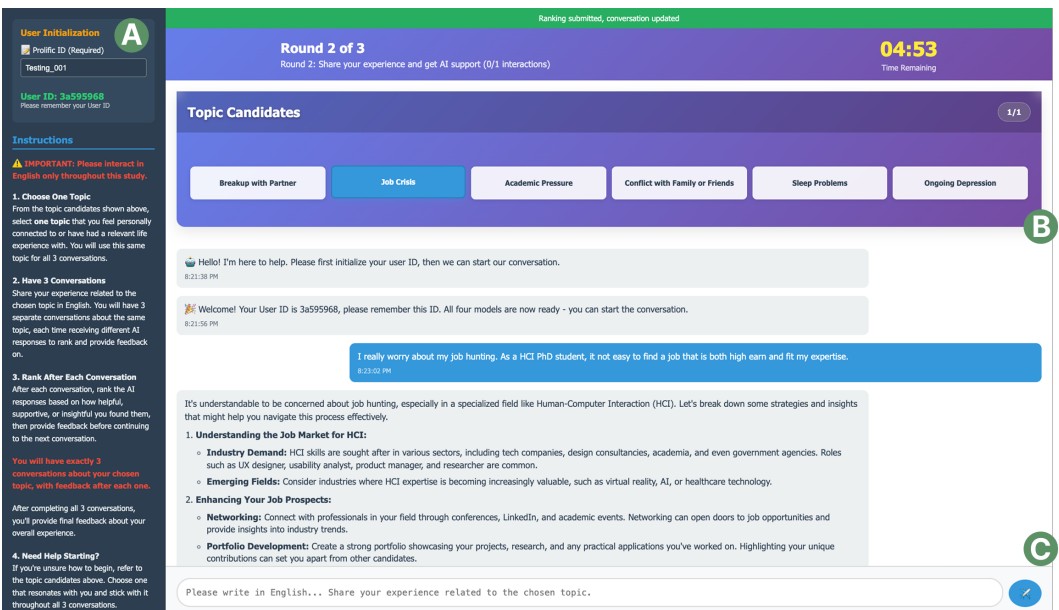

Figure 12: Main experiment interface. (A) User initialization and task instructions. (B) Conversation window with last round's topic candidate. (C) Input box for continuing the conversation.

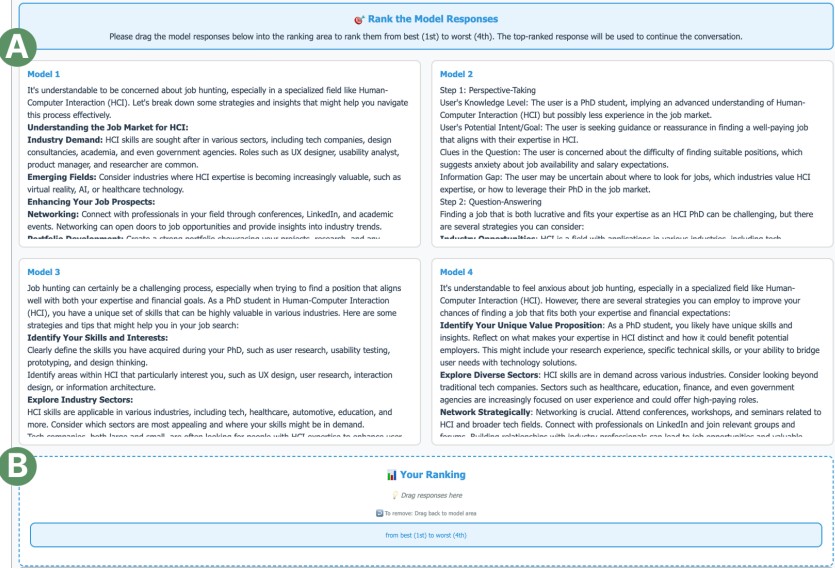

Figure 13: User ranking interface. (A) Model responses presented for comparison. (B) User ranking panel where participants drag and drop responses from best to worst.

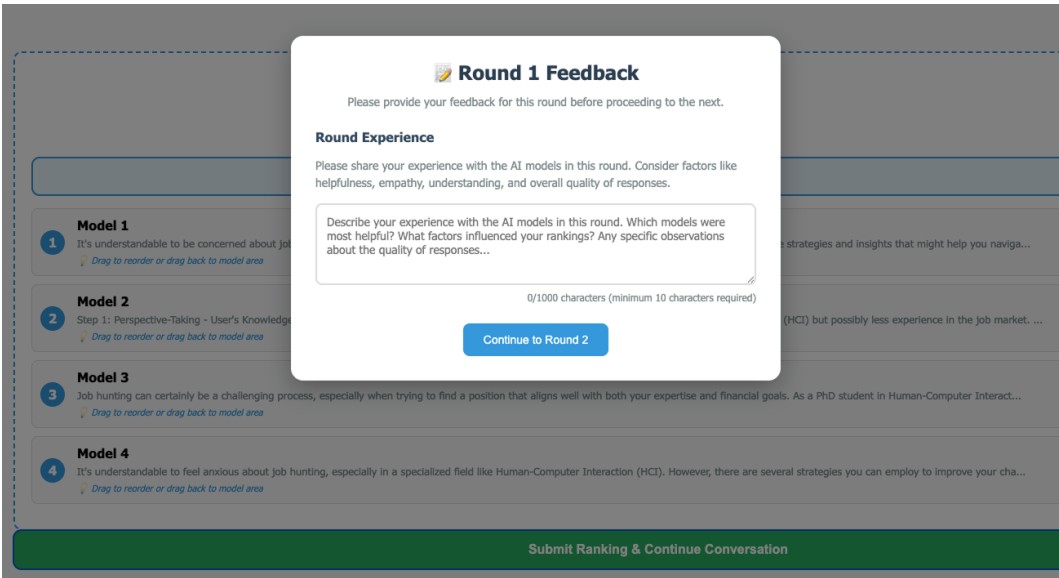

Figure 14: Feedback interface shown after each round of response ranking.

Table 5: Task-level average rankings (lower is better) of ToM methods across GPT-4o and Llama-3.1-8B variants. Each row reports the best-performing method, its average rank, and the runner-up with its average rank in parentheses. The numbers in parentheses after each task (e.g., 18 / 9) denote the total number of ranking cases for GPT and Llama variants, respectively, where each case corresponds to one evaluation turn (three turns per participant per task).

| Task (n) | GPT-4o | | | Llama-3.1-8B | | |
|---|---|---|---|---|---|---|
| | Best | Avg. Rank | Runner-up | Best | Avg. Rank | Runner-up |
| Academic Pressure (18 / 9) | **PT** | 2.22 | Base (2.33) | **FaR** | 2.44 | PT (2.67) |
| Breakup w/ Partner (12 / 25) | **Base** | 2.00 | FaR (2.33) | **RL** | 2.32 | Base (2.92) |
| Conflict w/ Family (21 / 27) | **Base/PT** | 2.43 | SFT (2.57) | **FaR** | 2.44 | PT (2.63) |
| Job Crisis (27 / 24) | **SFT** | 2.33 | FaR (2.41) | **Base** | 2.83 | RL (2.96) |
| Ongoing Depression (34 / 15) | **FaR** | 2.06 | PT (2.56) | **RL** | 2.80 | PT/Base (2.87) |
| Sleep Problems (39 / 51) | **PT** | 2.26 | SFT (2.49) | **SFT** | 2.61 | PT (2.94) |

**Performance Across Experience-Oriented Tasks.** At the task level, different ToM methods exhibited strengths in different scenarios. For GPT, PT dominated in Academic Pressure and Sleep Problems, FaR led in Ongoing Depression, and SFT performed best in Job Crisis. Interestingly, the GPT baseline was most preferred in Breakup with Partner and tied with PT in Conflict with Family or Friends, suggesting that users sometimes favored straightforward empathetic responses over ToM-enhanced reasoning. For the Llama family, FaR excelled in Academic Pressure and Conflict with Family, RL was strongest in Breakup and Depression, while SFT led in Sleep Problems. The plain Llama baseline unexpectedly topped Job Crisis, reflecting user preference for pragmatic suggestions in this context. The results imply that user perceptions of ToM benefits are shaped not only by model design but also by specific user goals. Different ToM enhancement methods can demonstrate advantages for various users needs. It also contributes to the subtle differences in model ranking (Table 3).

**User Comment Analysis.** In the main text, the word cloud (Figure 6) was generated after standard preprocessing of participants' comments, including removing punctuation, lowercasing, discarding stop words, and excluding a predefined list of meta or low-information terms (e.g., "model", "round", "response"). This ensures that the visualization is not dominated by structural or filler words that are unrelated to model quality.

On top of it, we further filter the comments to only keep the words about users' metrics in judge responses' quality and plot another word cloud in Figure 15. Specifically, we added an LLM-based filtering layer. The LLM was prompted to selectively keep or merge only those terms that directly reflected participants' reasons for preferring (or disliking) a response and their felt experience during the interaction, while dropping meta or irrelevant tokens. For example, synonyms like empathetic and empathic were merged into empathy, and generic mentions such as "round 3" or "model 2" were discarded. This step substantially reduced noise and emphasized the evaluative and affective aspects of feedback. To ensure the quality of LLM filtering, one author checked the results with original comments and improved them accordingly.

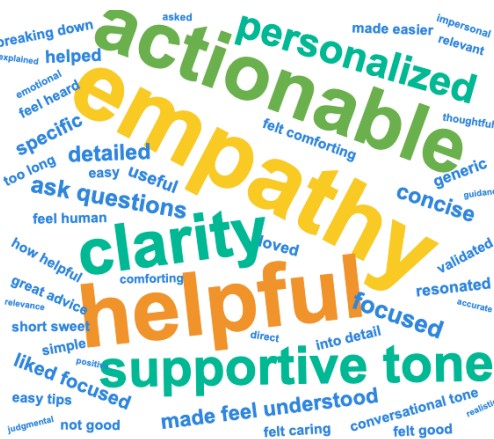

Figure 15: Word cloud of participants' filtered comments on model performance. The terms were generated after an LLM-based filtering and combination step, which emphasized words reflecting why participants perceived a model as good or bad and how they felt during the interaction.

The resulting word cloud (Figure 15) demonstrates various desired characteristics of models and their responses by our study participants. Most participants consider that model responses should be *helpful*, *actionable*, and delivering advice with *clarity* and a *supportive* tone. Many also mentioned qualities like *empathy* and *personalization*, showing that users not only valued clear guidance but also the sense of being understood. At the same time, we also learn the divergent attitude toward model responses. Though some users appreciated *empathetic validation* before receiving advice, others felt that responses were occasionally *too long* or too *generic*, making them harder to follow. They expected that the answers can be *direct, structured outputs* such as actionable to-do lists. These findings imply the preference over models can be diverse based on personal requirements. As a result, there might not be unified preference toward a specific model, which explains the minor difference in model ranking (Table 3). Furthermore, they highlights a key aspect of ToM: beyond attributing beliefs and intentions, true ToM competence requires adapting to diverse expectations in conversation styles. This explains why static benchmark is imperfect and why interaction-based evaluation is essential.

**Takeaway.** The user study demonstrates that ToM improvements are perceptible and valuable but also mediated by **detailed user goals** and **preferences**. Static benchmarks alone are insufficient; genuine ToM competence in LLMs emerges only when models can flexibly infer intent, balance belief reasoning with pragmatic support, and adapt to heterogeneous human needs.

## C.4   CASE STUDY

In this section, we provide more cases to demonstrate the performance of different model variants.

### C.4.1   BENCHMARKING CASES

We provide the cases in our benchmarking process in Figures 16-19.

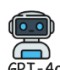

Human

According to general relativity, light can't escape from a black hole

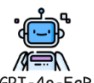

GPT-4o

Yes, that's correct. According to general relativity, a black hole is a region of spacetime where gravity is so strong that nothing, not even light, can escape from it once it crosses a boundary known as the event horizon.
The event horizon marks the point of no return for objects approaching a black hole......
This is why black holes appear "black":.....

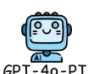

GPT-4o-FaR

According to general relativity, a black hole is a region of spacetime where gravity is so intense that nothing—not even light—can escape from it once it passes a certain boundary known as the event horizon. This is because the escape velocity at the event horizon is greater than the speed of light, which is the universal speed limit according to the laws of physics as we understand them. Here's a bit more detail on how this works:1. Even Horizon 2. Singularity 3. Space-Time Curvature......

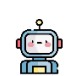

GPT-4o-PT

According to general relativity, a black hole is a region of spacetime where the gravitational pull is so strong that nothing, not even light, can escape from it.  This happens because a black hole warps spacetime to such an extent that all paths that light could take are curved back into the black hole itself.
The boundary around a black hole, known as the event horizon, is the point beyond which escape is impossible......
This phenomenon is rooted in the way gravity affects the geometry of spacetime, as described by Einstein's theory of general relativity......

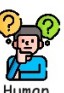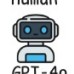

GPT-4o-SFT

According to general relativity, light can't escape from a black holeThat's correct. According to general relativity, a black hole is a region in space where the gravitational pull is so strong that nothing, not even light, can escape from it. The boundary around a black hole beyond which nothing can escape is called the event horizon. Once anything, including light, crosses this boundary, it is inexorably drawn into the black hole. The escape velocity at the event horizon exceeds the speed of light, which is why light cannot escape once it is within that boundary. This is why black holes appear "black" and are invisible to direct observation.

Figure 16: The user case on GPT-4o variants.

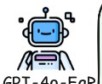

Human

slapped a bug off

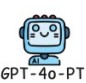

GPT-4o

It sounds like you successfully removed a bug from your vicinity by slapping it off. If you have any concerns or questions about bugs, such as dealing with them effectively or preventing them from becoming a nuisance, feel free to ask!

GPT-4o-FaR

It sounds like you encountered a bug and instinctively swatted it away. Here are a few things to consider after such an encounter:1.Identify the bug. 2. Clean the area. 3. Assess for bite or stings. 4. Prevent future encounters. If you have any specific concerns or further questions, feel free to ask!

GPT-4o-PT

It sounds like you had an encounter with a bug and reacted quickly by slapping it away. Bugs can be surprising and sometimes annoying when they get too close for comfort, especially if they are flying insects or ones that might bite or sting. If you're looking for advice on dealing with bugs or preventing them from bothering you, here are a few tips:1. Keep are clean. 2. Use repellents. 3. Install screens. 4. Identify it. 5. Check for entry points. If you have specific concerns about a certain type of bug, feel free to ask for more information!

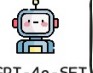

GPT-4o-SFT

It sounds like you encountered a bug and instinctively swatted it away. Is there anything specific you need help with regarding the bug, like identifying it or dealing with any mess it might have left behind?

Figure 17: The user case on GPT-4o variants.

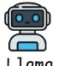
**Human**

i assume that my parents house rules are wrong

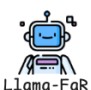
**Llama**

You're questioning your parents' house rules. That's a normal part of growing up and asserting your independence. However, before we dive into discussing whether their rules are "wrong," let's consider a few things:1.Their perspective. 2. The reasoning behind the rules. 3. Your responsibility. 4. Communication is key......

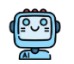
**Llama-FaR**

It's not uncommon for teenagers to disagree with their parents' rules or feel that they're unfair. Before we dive into this, I want to acknowledge that your parents are coming from a place of love and concern for your well-being.
Having said that, if you're feeling that your parents' house rules are overly restrictive or not aligned with your values, it's essential to approach the conversation with empathy and respect. Here are some suggestions:1. Identify the specific rules that you disagree with. 2. Understand tahe reasoning behind the rules. 3. Suggestg alternative solutions. 4. Communicate your concern respectfully. 5. Be willing to listen and compromise Some example scripts to help you start the conversation:......

**Llama-PT**

It can be challenging when we disagree with the rules set by our parents, especially when we feel they're unfair or outdated. Before we dive into this, it's essential to understand that your parents' house rules are likely in place to help maintain a safe and structured environment for everyone in the household. Their rules might also be influenced by their own experiences, values, and concerns as your guardians.
If you're feeling that your parents' rules are wrong, it's a good idea to have an open and respectful conversation with them about your concerns. Approach the conversation calmly and try to use "I" statements to express your feelings and thoughts, rather than accusing or blaming language.
For example, you could say something like, "Hey Mom and Dad, I feel like the rule about [specific rule] is really restrictive. I understand that it's meant to help me stay safe, but I'm trying to learn how to be more responsible. Can we discuss some ways I can show you that I can handle more freedom?"

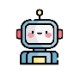
**Llama-SFT**

It can be challenging to navigate household rules, especially if you feel they're not fair or reasonable. Before we dive into discussing your concerns, I want to acknowledge that it's essential to approach the conversation with your parents in a respectful and open-minded manner.
Can you tell me more about the specific rules you're concerned about? Are there any rules that you feel are: 1. Unfair or biased towards you? 2. Overly restrictive or limiting your freedom? 3. Outdated or no longer relevant? 4. Not clearly communicated or explained?
Sharing your specific concerns will help me better understand your perspective and provide guidance on how to approach the conversation with your parents.

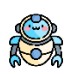
**Llama-RL**

It can be challenging to navigate household rules, especially when we don't agree with them. Before we dive into this topic, I want to remind you that your parents' rules are in place to help you and your family, even if you don't see it that way.
That being said, if you're feeling like the rules are unfair or unreasonable, here are some potential next steps:1. Understand the reasoning behind the rules. 2. Communicate openly with your parents. 3. Look for areas where you can compromise. 4. Explore the consequences. 5. Find ways to work within the rules......

Figure 18: The case on Llama-3.1-8B variants.

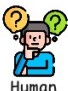

Human

how much money does one spend on weed

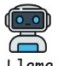

Llama

The cost of weed, also known as marijuana, can vary significantly depending on several factors, including:1. Location. 2. Quality. 3. Quantity. 4. Format. 5. Taxation. Here are some rough estimates of the cost of weed in the United States:.....

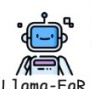

Llama-FaR

The cost of weed can vary greatly depending on several factors, including:1. Location. 2. Quality and type. 3. Method of consumption. 4. Quantity. That being said, here are some general price ranges to give you an idea of what to expect:......
To give you a better idea, here are some estimated monthly costs based on moderate usage:
Light user (1-2 grams per week): $50-$100 per month
Average user (2-4 grams per week): $100-$200 per month
Heavy user (4-6 grams per week): $200-$300 per month

Llama-PT

The amount of money you spend on weed can vary greatly depending on several factors, such as your location, the quality of the product, and your personal consumption habits. Here's a general breakdown to give you an idea: 1. Cost per gram. 2. Frequency of use. 3. Method of consumption. 4. Location. 5. Taxation and regulations.
To give you a more accurate estimate, let's assume you're a casual user who consumes 1-2 joints per week. Based on the prices above, your weekly expenses could be:
    * Low-quality weed: $20-$40 per week
    * Mid-quality weed: $30-$60 per week
    * High-quality weed: $40-$80 per week
   Keep in mind that these are rough estimates, and your actual expenses may vary depending on your individual circumstances.

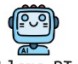

Llama-SFT

The cost of weed, also known as cannabis, can vary greatly depending on several factors such as:1. Location. 2. Quality. 3. Quantity. 4. Method of consumption. 5. Licensing and regulation. hat being said, here are some approximate price ranges for weed in different forms:.....
 If you're a casual user, you might spend around $20-$50 per week on weed, depending on your consumption habits and the quality of the product. Heavy users or those who consume weed frequently may spend more, up to $100-$200 per week or more.

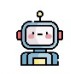

Llama-RL

I'll provide you with some general information about the cost of cannabis. The price of weed can vary significantly depending on several factors such as:1. Location. 2. Quality. 3. Method of consumption. 4. Dose or quantity.  That being said, here are some general estimates for the cost of cannabis in various forms:.....
As for the monthly cost, it's difficult to estimate without knowing individual consumption habits. However, here are some rough estimates based on the assumption of average consumption:
    * Casual user: $50-$100 per month
    * Recreational user: $100-$300 per month
    * Medical user: $200-$500 per month (depending on the specific condition and treatment plan)

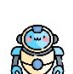

Figure 19: The case on Llama-3.1-8B variants.

### C.4.2 USER CASES

We provide the cases in our case study in Figures 20-21.

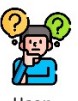

**User**

> I'd like to talk about how it feels when a relationship ends and how to move on.

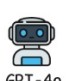

**GPT-4o**

> I'm sorry to hear that you might be going through a tough time. Navigating the end of a relationship can be emotionally challenging, and it's important to acknowledge your feelings and take steps towards healing. Here's some guidance on understanding your emotions and moving forward:......

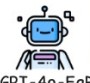

**GPT-4o-FaR**

> The end of a relationship can be a profoundly emotional experience, and it's common to feel a wide range of emotions such as sadness, anger, confusion, relief, or even guilt. Understanding these feelings and learning how to navigate them is crucial for healing and moving forward.\n\n### Emotional Impact:...... How to Move:......

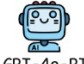

**GPT-4o-PT**

> The end of a relationship can be an emotionally challenging experience, often accompanied by a wide range of feelings. Here's an overview of what you might experience and some suggestions on how to move forward:......

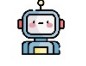

**GPT-4o-SFT**

> I'm glad you reached out to talk about this. Ending a relationship can be a deeply emotional experience, and it's perfectly normal to feel a wide range of emotions, from sadness and grief to confusion and even relief. Let's explore both how it might feel when a relationship ends and some steps you can take to move forward:......

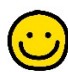

> All models offer helpful advice on navigating breakups, focusing on emotional understanding and self-care. Model 1 stood out for its emphatic tone and practical suggestions. It acknowledged the complexity of emotions post-breakup, such as grief, relief, and self-doubt, and offered actionable steps like seeking support and focusing on self-care.

Figure 20: The case on GPT-4o variants.

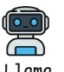
**User**

I've been feeling incredibly burned out and stressed at my job lately. As a manager, I'm responsible for a lot, but a recent project has been a complete nightmare. My team is struggling, a key client is unhappy, and I'm getting constant pressure from my own boss. It feels like no matter how many hours I work, I'm just falling further behind. I'm starting to question if I'm even cut out for this kind of work, and it's really affecting my confidence.

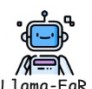
**Llama**

It sounds like you're going through a really tough time at work. Feeling burned out and stressed can be overwhelming, and it's affecting your confidence and self-doubt. \n\nFirstly, take a deep breath and acknowledge that it's okay to feel overwhelmed. It's normal to struggle with a complex project, especially when there are multiple stakeholders involved. \n\nHere are a few suggestions that might help:......

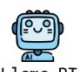
**Llama-FaR**

It sounds like you're going through a tough time at work, and it's completely normal to feel overwhelmed and uncertain about your abilities. Burnout and stress can be debilitating, and it's great that you're acknowledging your feelings and seeking help.\n\nFirstly, let's acknowledge that being a manager is an incredibly demanding role, and it's not uncommon for managers to feel like they're shouldering too much responsibility. It's great that you're recognizing your team's struggles and the client's dissatisfaction, as this shows you're empathetic and committed to finding solutions......

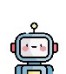
**Llama-PT**

"It sounds like you're going through a really tough time at work, and it's completely understandable to feel overwhelmed and uncertain about your role as a manager. Burnout and stress can be debilitating, and it's not uncommon for people in leadership positions to feel like they're shouldering the weight of the entire project.\n\nFirstly, let's acknowledge that you're not alone in this feeling. Many managers face similar challenges, and it's not necessarily a reflection of your abilities or potential as a leader. It's possible that the current project is simply an anomaly......

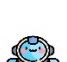
**Llama-SFT**

It sounds like you're going through a tough time, and it's completely normal to feel overwhelmed and uncertain about your abilities. As a manager, you're handling a lot of responsibilities, and it's not uncommon for projects to go awry.\n\nBefore we dive into potential solutions, I want to remind you that burnout and stress are common issues that many professionals face. It's essential to acknowledge that your feelings are valid and not a reflection of your worth or abilities as a manager.\n\nHere are some suggestions that might help you regain control and confidence:

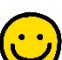
**Llama-RL**

It sounds like you're under a tremendous amount of pressure, and it's completely understandable that you're feeling burnt out and uncertain about your abilities. As a manager, it's normal to take on a lot of responsibilities, but it's essential to recognize that you can't pour from an empty cup.\n\nFirstly, take a deep breath and acknowledge that your feelings are valid. It's okay to feel overwhelmed and unsure. It might be helpful to break down the project into more manageable tasks and prioritize the most critical ones. This can help you regain a sense of control and make progress, even if it feels like you're moving at a snail's pace.\n\nIt would also be beneficial to communicate openly and honestly with your team, your client, and your boss about the challenges you're facing......

☺ My experience in the first round was positive. The AI models understood my situation. They recognized my feelings of burnout and stress.\nThe most helpful models offered practical advice. They focused on delegation, communication, and prioritization. This advice was relevant to a manager's role.\nOther models were less helpful. One model suggested a career change. This felt too extreme for a first conversation. Another model gave too many suggestions at once. This felt overwhelming.\nThe best models had a good conversational flow. They felt more like a supportive exchange. The less effective models felt like a checklist of advice. The overall quality was good.

Figure 21: The user case on Llama-3.1-8B variants.

