# OpenReview forum: "From Assistants to Companions: Towards the Usefulness of Improving Theory of Mind for Human-AI Symbiosis"
_ICLR.cc/2026/Conference — ICLR 2026 Conference Withdrawn Submission_

### Official Review · Reviewer_DeuF · 2025-10-26

**Soundness:** 3
**Presentation:** 2
**Contribution:** 3
**Rating:** 4
**Confidence:** 3

**Summary:**

This paper introduces an evaluation framework for LLMs' ToM in the human-AI (HAI) context, shifting from static third-person story-acting to dynamic engagement. The authors assess LLM performance on two interaction scenarios (goal-oriented and experience-oriented) via existing benchmarks and a crowdsourcing user study (N=100). The experiments reveal that static benchmark improvements do not always translate to better dynamic HAI interaction performance.

**Strengths:**

1. The challenges and motivation proposed in this paper are clear, promising, and interesting. I like the problem revealed: "the existing benchmarks for Large Language Models (LLMs) focus on testing their ToM capability with story-reading from a third-person perspective, leading to a critical gap between benchmark performance and practical competence in HAI collaborative and supportive tasks."
2. The problem shifting from test-based assessments to dynamic, real-world interaction challenges is intuitive and interesting.
3. This paper propose a comprehensive benchmarking framework and conduct a user study to support the findings.

**Weaknesses:**

1. Although I acknowledge the limitation of previous ToM benchmarking work stated in this paper (third-person, story acting), I have doubts about using **code and math** tasks to test ToM. These two tasks seem to have a **weak connection to ToM**.
2. The authors propose a framework, but they use previous data for evaluation instead of constructing their own, which weakens the contribution. I believe the **user study** (N=100) is a more significant contribution, yet the paper devotes insufficient space and analysis to it. Therefore, the focus of this paper may require some reorganization.
3. The performance gap between ToM and the tasks in this paper, from the perspective of deep learning, seems to stem more from the **inconsistency between training objectives and downstream tasks**. The paper's analysis remains at a **superficial**, result-oriented level (both quantitative and qualitative), lacking in - depth analysis of intermediate processes. For instance, it fails to explore the **specific role of ToM** when LLMs perform various tasks, as well as the explicit connection between ToM and these specific tasks.

**Questions:**

1. Are there any target-oriented tasks that require collaborative LLMs to be aware of each other’s thoughts (ToM)?
2. Do the tasks evaluated in the paper explicitly or implicitly use any results or intermediate steps enhanced by ToM? This is crucial, if not, the performance will naturally not improve (target-oriented tasks); if yes, it can explain why the improvement occurs (experience-oriented tasks).

---

> ### Author Response · Authors · 2025-12-02
>
> Thanks for your comments.
>
> ## For W1
> On the one hand, goal-oriented tasks (e.g., math, code) are important in HAI scenarios. Our benchmarking goal is to investigate if the ToM can improve HAI symbiosis, so we need to include this part. On the other hand, even in these problem-solving processes, it contains mental states such as belief, intention, and knowledge. We assume an AI with excellent ToM can correctly answer, even in cases that the questions are ambiguous.
>
> ## For W2
> Although the data is from the existing works, we carefully design the comprehensive evaluation pipeline including simulating both the collaboration and interaction between human and AI.
>
>
> ## For W3
> The point “inconsistency between training objectives and downstreams tasks” in ToM is what we try to reveal. Current ToM training is based on the goal to improve the performance on the static benchmarks. However, the performance of question answering doesn’t represent the real-world ToM ability. In other words, a well defined ToM benchmark should be designed to reflect how good it can understand users' mental states and help users accordingly. However, our study demonstrates the misalignment between the current ToM benchmark and the real goal of increasing ToM of LLMs. We believe the well-designed ToM-enhancing method should directly make a difference on the users.
>
> ## For Q1
> We believe even in the goal-oriented tasks, ToM is needed. And our study reveals the difference between LLMs with different ToM levels. For example, in Figure 5 (left), we provide an example of goal-oriented tasks. LLMs with ToM can understand more what the users try to seek. We believe the taxonomy such as belief, intention, and knowledge is also important in these tasks.
>
> ## For Q2
> Yes, ToM reasoning is utilized either explicitly or implicitly, directly explaining the performance divergence: Explicit (Prompting): Methods like FaR and PT force the generation of intermediate social reasoning steps. Implicit (Fine-Tuning): SFT and RL embed the social knowledge into the model weights.

---

### Official Review · Reviewer_Y1BR · 2025-10-31

**Soundness:** 1
**Presentation:** 2
**Contribution:** 1
**Rating:** 2
**Confidence:** 4

**Summary:**

The paper proposes a framework to evaluate LLMs’ Theory of Mind (ToM) capabilities in dynamic, first-person human–AI interactions. It claims to cover both goal-oriented tasks (e.g., coding, math) and experience-oriented tasks (e.g., counseling), aiming to bridge the gap between traditional third-person story-based benchmarks and real-world interactive scenarios. The framework includes four synthesized benchmarks and a crowdsourcing user study to assess LLM performance.

However, in practice, the proposed evaluation is largely superficial: it consists mainly of trivial modifications to existing datasets, such as changing character names or narrative perspective, without actually modeling or analyzing any mental states. The evaluation methodology itself is not grounded in ToM reasoning, and thus the framework does not meaningfully capture the dynamic, interactive capabilities that the paper claims to address. Essentially, the study applies a ToM label to out-of-domain tasks rather than genuinely evaluating dynamic ToM in human–AI interaction.

**Strengths:**

1. Figures are pretty.

**Weaknesses:**

1. It is unclear what type of ToM the authors intend to evaluate, and the term “dynamic” is not formally defined. The only apparent modification is a shift in narrative perspective (first-person vs third-person), which does not constitute genuine dynamic or interactive reasoning. Evaluation metrics are largely task-dependent rather than analyzing any mental states, raising doubts about whether the methodology truly reflects dynamic, interactive ToM.
2. Evaluating ToM on math and code generation tasks appears unnecessary, as these tasks primarily require formal reasoning and problem-solving rather than modeling mental states or intentions. The chat-like format does not involve genuine interaction, suggesting a limited understanding of ToM and undermining the relevance of the results. The results in Table 1 prove this.
3. The three contributions listed by the authors are not clearly reflected in the experiments and the method design.
4. Experiments focus mainly on task performance, but it is unclear whether the LLMs actually perform ToM reasoning, since none of the ToM components are explicitly analyzed.
5. The study does not include an experiment comparing the effectiveness of “dynamic” versus “static” ToM in HAI interactions, as claimed in the abstract.
6. Ambiguity in tables: The meaning of values in Table 1 and Table 2 is unclear—are they 𝑀Γ(𝜋𝐴, 𝑇) scores or task success rates?
7. Minor typos: Line 144: “a active participant” → “an active participant.” Line 145: “from the first-person perspective” → “from a first-person perspective.”

**Questions:**

1. What is the meaning of the values in Tables 1 & 2? And how are these numbers computed?

---

> ### Author Response · Authors · 2025-12-02
>
> Thanks for your comments.
> ## For W1
> The “dynamic” is reflected in the interaction loop between humans and AI. We simulate multi-turn conversation between human and AI to implement a dynamic evaluation which captures the real ToM ability in HAI interaction. Since our goal is to study the effectiveness of ToM, the metrics we used aim to reflect the performance on the specific tasks.
>
> ## For W2
> We regard goal-oriented tasks (e.g., code, math) as an important part, because they appear frequently when we interact with AI. Even in the problem-solving process, it may contain mental states such as intention, belief, and knowledge. If the AI understands humans, it will perform the instructions, even if it is ambiguous, the AI can understand and answer correctly.
>
>
> ## For W3
> For the first contribution “problem formulation”, we frame the ToM problem in HAI setting in Sec. 2.2. For the second contribution “evaluation on goal- and experience-oriented tasks and user study”, we include all the experiments and user study in Sec. 4. For the third contribution “reveal limitations and offer insights”, we have detailed discussion in Sec. 5.
>
> ## For W4
> We have provided a pre-testing to study if the testing methods can improve ToM performance in Sec. 3.1. The results demonstrate they all work on the ToM filed. Since our goal is to study the effectiveness of ToM, we mainly investigate the downstream performance, i.e., task performance, to see if we really need ToM in HAI interaction.
>
>
> ## For W5
> To clarify, we compare the “static” ToM in story-question-option paradigm and “dynamic” HAI interaction. We have included a “static” evaluation sample in Sec. 3.1. It shows all the methods perform well on it. However, they fail to show clear improvement on the real-world HAI interaction.
>
> ## For W6
> We have elaborated on the metrics used in different datasets, please refer to Sec. 4.1 and Sec. 4.2. For Table 1, the values are the accuracy of the final answers. For Table 2, the values are ratings from LLM-as-a-judge.
>
> ## For W7
> Thanks for pointing it out, we will correct them.
>
> ## For Q1
> We have elaborated on the metrics used in different datasets, please refer to Sec. 4.1 and Sec. 4.2. For Table 1, the values are the accuracy of the final answers. For Table 2, the values are ratings from LLM-as-a-judge.

---

### Official Review · Reviewer_9NNL · 2025-11-01

**Soundness:** 2
**Presentation:** 3
**Contribution:** 2
**Rating:** 2
**Confidence:** 2

**Summary:**

This paper raises the important point that there exists a significant gap between benchmark performance and practical competence in Human-AI (HAI) collaboration, and proposes an evaluation protocol that shifts from third-person story-based assessment to first-person interactive evaluation. The authors introduce a novel framework that categorizes HAI scenarios into goal-oriented tasks (e.g., coding, math problem-solving) and experience-oriented tasks (e.g., counseling, emotional support) and systematically evaluate four existing ToM enhancement methods (FaR, PT, SFT, RL) across two model families (GPT-4o and Llama-3.1-8B) using both synthetic benchmarks and a crowdsourced user study with 100 participants. Key findings reveal a significant performance gap: improvements on static ToM benchmarks do not translate to better performance in dynamic HAI interactions, demonstrating the necessity of interaction-based assessment and revealing that current methods fail in goal-oriented tasks while showing modest gains in experience-oriented scenarios.

Overall, this is an interesting paper addressing an important hypothesis. However, it feels more suited for a findings paper rather than a main conference contribution, as the primary contribution is empirical validation of intuitive concerns about benchmark-reality gaps.

**Strengths:**

- **Strong motivation and intuition:** Addresses the fundamental problem of static benchmark limitations by proposing a shift to first-person perspective evaluation, which better reflects real-world HAI interactions where models must understand users' mental states dynamically.
- **Well-designed experimental framework:** The experiments are systematically conducted under the proposed framework, with the authors validating results through real human user studies involving 100 participants across six experience-oriented tasks, which is crucial for a HAI symbiosis topic where human perception matters.
- **Methodological rigor:** The authors clearly formalize the evaluation shift through mathematical notation (Equations 1-4), systematically adapt existing ToM methods for first-person scenarios, and provide detailed statistical analysis across multiple benchmarks, establishing a solid foundation for reproducible evaluation.

**Weaknesses:**

- **Limited novelty and scope:** The contribution is primarily empirical evaluation under the proposed framework rather than methodological innovation, making it more suitable for a findings paper.
- **Oversimplified methodology adaptation:** The conversion from third-person to first-person scenarios relies on simple rule-based transformations (e.g., replacing protagonist names with "I"), which seems to maintain the fundamental structure of static scenarios and not well-justified for capturing the complexities of real world, first-person-perspective conversations

**Questions:**

- Given that the third-person to first-person conversion is rule-based and static, does this transformation still capture the dynamic nature of real HAI interactions that the authors claim to evaluate? The conversion seems to maintain the fundamental structure of static scenarios.
- The paper mentions "the top-ranked response is used to continue the dialogue" - this means different models could be used in different turns within a single dialogue. How does this multi-model approach affect the validity of model-specific performance evaluation, and how do you isolate individual model contributions?
- The conclusion that "limited gains are not strongly perceived by users, failing to translate into a clear preference" needs better quantification. Is this based on the lack of statistical significance in ranking differences, the small effect sizes observed in Table 3, or the qualitative feedback analysis? The paper would benefit from clearer statistical thresholds and effect size reporting for these claims.

---

> ### Author Response · Authors · 2025-12-02
>
> Thanks for your comments.
> ## For W1
> We respectfully argue that our contribution extends beyond standard empirical evaluation. We propose a fundamental methodological shift in how Machine ToM is conceptualized and measured, moving from static, isolated benchmarks to dynamic, symbiotic HAI interactions.
>
> While we do not propose a new model architecture, the proposed HAI-based evaluation framework itself is a methodological innovation. Current static benchmarks have become saturated or misaligned with real-world utility. Our framework fills this critical gap by establishing a new standard for assessing ToM, which serves as a prerequisite for developing the next generation of socially intelligent AI. Therefore, this work provides the necessary infrastructure and methodology for future model development, fitting the scope of a main track contribution.
>
> ## For W2
> Our adaptation only wants to maintain the overall ideas of these methods and make it suitable for first-perspective benchmarking. Our results on HiToM-first show that they can achieve great performance on the static benchmarks, but fail on real-world tests. Our goal is benchmarking rather than building new methods.
>
> ## For Q1
> The conversion doesn’t change the nature of the methods. It only enables the methods to be suitable to be evaluated by our first-perspective scenarios. We further benchmark the methods to see if they can be used on dynamic real HAI interactions.
>
> ## For Q2
> Because we focus on turn-level evaluation, the context doesn’t affect the fairness of evaluation on different models in a single turn.
>
> ## For Q3
> We have provided detailed statistical significance tests in supplementary materials.

---

### Official Review · Reviewer_1ry9 · 2025-11-01

**Soundness:** 3
**Presentation:** 3
**Contribution:** 3
**Rating:** 4
**Confidence:** 3

**Summary:**

It proposes a dynamic, first-person evaluation framework that reveals how improvements on static Theory of Mind benchmarks for LLMs do not necessarily translate to better real-world human-AI performance, highlighting the need for interaction-based assessment and socially aware models.

**Strengths:**

- The paper is well-written and well-motivated. It points to the importance of ToM in human-AI interaction experience.

- The paper highlights the importance of first-person view tasks. It includes both goal-oriented and experience-oriented scenarios.

- It shifts the evaluation from static to dynamic interaction setting.

**Weaknesses:**

- The main weakness of this paper is the test data is from existing benchmarks which lower the weight of the contribution.

- The GPT-4o is a little bit old, the author should add more SOTA models into the evaluation.

- In my view, ToM is about the cognitive facets, why do you conduct the experiments on goal-oriented tasks? What problem would you like to study?

- There lacks in-depth analysis about why SFT and RL methods do not show the advantages on the tested benchmarks in Table 2. Currently, more description about the results scores in Section 4.2.1 and 4.2.2, it should have more analysis.

**Questions:**

See weakness.

---

> ### Author Response · Authors · 2025-12-02
>
> Thanks for your comments.
> ## For W1
> The main contribution of our paper is to shift the evaluation of ToM from model-centric to human-centric [1]. We formulate the problem of ToM in HAI symbiosis and construct a comprehensive evaluation framework to study the effectiveness of ToM-enhancing methods on different HAI scenarios.
>
> [1] Liao, Q. Vera, and Ziang Xiao. "Rethinking model evaluation as narrowing the socio-technical gap." arXiv preprint arXiv:2306.03100 (2023).
>
>
> ## For W2
> Our goal is to study the effectiveness of ToM-enhancing methods rather than the base models. Therefore, we involve as many as possible methods on two widely-used base models.
>
>
> ## For W3
> Our evaluation goal is HAI symbiosis. Goal-oriented tasks are an important part in HAI symbiosis. Therefore, we want to investigate if ToM, the cognitive facets, can make a difference on goal-oriented tasks.
>
> ## For W4
> In Sec 5, we discuss the insights behind the performance. We think the generalizability of the methods is the main problem. Trained on the static story-question-option datasets, the LLMs don’t learn real ToM needed for HAI symbiosis.

---

### Note · Authors · 2026-01-06

I have read and agree with the venue's withdrawal policy on behalf of myself and my co-authors.